# Effect of Various Forms of Aluminum 6082 on the Mechanical Properties, Microstructure and Surface Modification of the Profile after Extrusion Process

**DOI:** 10.3390/ma14175066

**Published:** 2021-09-04

**Authors:** Piotr Noga, Andrzej Piotrowicz, Tomasz Skrzekut, Adam Zwoliński, Paweł Strzępek

**Affiliations:** Faculty of Non-Ferrous Metals, AGH University of Science and Technology, A. Mickiewicz Av. 30, 30-059 Cracow, Poland; andpio@agh.edu.pl (A.P.); skrzekut@agh.edu.pl (T.S.); zwolo@agh.edu.pl (A.Z.); strzepek@agh.edu.pl (P.S.)

**Keywords:** recycling, 6082 aluminum alloy, plastic consolidation, mechanical properties, microstructure, surface modification

## Abstract

This article presents a method of reusing aluminum scrap from alloy 6082 using the hot extrusion process. Aluminum chips from milling and turning processes, having different sizes and morphologies, were cold pressed into briquettes prior to hot pressing at 400 °C at a ram speed of 2 mm/s. The study of mechanical properties combined with observations of the microstructures, as well as tests of density, hardness and electrical conductivity were carried out. On the basis of the results, the possibility of using the plastic consolidation method and obtaining materials with similar to a solid ingot mechanical properties, density and electrical conductivity was proven. The possibility of modifying the surface of consolidated aluminum scrap was tested in processes examples: polishing, anodizing and coloring. For this purpose, a number of analyses and tests were carried out: comparison of colors on color histograms, roughness determination, SEM and chemical composition analysis. It has been proven there are differences in the surface treatment of the solid material and that of scrap consolidation, and as such, these differences may significantly affect the final quality.

## 1. Introduction

Recycling of light metals is largely based on the remelting method. For this purpose, various types of melting furnaces are used, from low-cost gas furnaces to electric furnaces with a special system of loading the initial material into the liquid metal which significantly increases the recycling yield, but also increases the processing cost. Recycled metal is subjected to the treatment processes: refining, chemical composition correction and degassing, and then casting into ingots. The average recycling yield when melting such scrap metal ranges from 75 to 81%. The low recycling yield of the melting methods (especially small chips, tapes, thin wires) causes irreversible losses of metal (forms low-quality slags and drosses) and energy used for the aluminum production [1,2,3,4,5,6].

The concept of recycling based on the plastic consolidation method creates opportunities to reduce losses during recycling and significantly reduces the energy consumption. This process completely omits the melting phase, thus eliminating losses and troublesome waste (drosses). The entire recycling process takes place in the solid phase, which avoids gasification effects of the liquid metal and loss of alloy additives during the metallurgical synthesis. The upsides of the process are lowering of the process temperature to approx. 350–450 °C, shortening the operation time to a few minutes and reducing energy consumption for reheating of the charge material several times. It also eliminates the risk of material losses as a result of oxidation, as surface oxidation is limited in this temperature range by a tight passivation layer [7,8].

Extrusion of fine fractions, such as metal chip, dusts, powders, tapes, etc., allows an effective plastic consolidation and the obtaining of a coherent material which has good utility properties [9,10,11,12,13,14,15]. Consolidation of the 6082 alloy chips by pressing and extrusion makes it possible to obtain briquettes with better fatigue properties compared to the solid material [16,17,18]. One of the materials most often used in the industry is aluminum and its alloys, in particular, alloy 6082. It is characterized by good resistance to corrosion and cracking. An interesting approach to the use of recycling by the extrusion method was applied to the AlSi11 alloy chips. After the plastic consolidation process, the obtained flat bars are subsequently welded, which leads to the production of metallic foams [19]. It is susceptible to machining and welding; it has decent mechanical strength. The above-mentioned properties enable the use of this material, among others, in the shipbuilding and automotive industries, where one has to deal with a corrosive environment and where heavy loads are transferred [20,21].

Aluminum is an amphoteric element and can dissolve both in an acidic and an alkaline environment, to form Al^3+^ and H_2_AlO_3_^−^, respectively. In the pH range from 4.45 to 8.38, aluminum is covered with durable oxide layer of Al_2_O_3_∙H_2_O (bemite). The formation of a thick protective coating is possible in a specially directed corrosion process in oxidizing environments, both acidic and alkaline, with pH values outside the above-mentioned area. Oxide layer formatted on aluminum or its alloys by electrolytic methods is characterized by some special properties such as protection against weakly aggressive environments [22,23].

One of the ways to improve the surface properties of metals is to produce the so-called conversion coatings. One of them is oxide coatings, which are produced by an electrolysis process called anodizing. In the case of aluminum, aluminum oxide films are obtained in the anodizing process carried out in an electrolysis cell, in which the anode is an oxidized aluminum, while the cathode is an electrolyte-resistant metal (e.g., lead). During anodic oxidation of aluminum, the anode reaction is described by the summary equation:(1)$$2{\mathrm{Al}}_{\mathrm{solid}}+3H_{2}O_{\mathrm{solution}}\to {\mathrm{Al}}_{2}O_{3}{}_{\mathrm{solid}}+6H^{+}+6e^{-}$$

The aluminum oxide layer is formed as a result of the reaction of Al^3+^ ions (formed by the direct current) with O^2−^ ions or OH^−^ formed by the decomposition of water. Evolution of hydrogen gas takes place at the cathode:(2)$$2H^{+}+2e^{-}\to H_{2}{}_{\mathrm{gas}}↑$$

Aluminum oxide layer is composed of densely arranged hexagonal cells with a pore inside. Due to the characteristic structure of the oxide layer (with relatively regular and densely spaced pores), the aluminum anodized surface is highly developed and has decent adsorption capacity. It is used in the coloring process to protect and decorate aluminum products like aluminum components [23,24].

Surface treatment with anodizing process includes a series of technological steps: 1. Surface preparation, 2. Anodizing, 3. Coloring, 4. Sealing. Preparation of aluminum for anodizing is extremely important because it determines whether anodizing will be successful. Firstly, the surface of the aluminum should be degreased in organic solvents, such as ethanol, acetone or carbon tetrachloride, to remove organic impurities and improve electrolyte wetting. If a special surface appearance (e.g., high gloss) is required, then mechanical or chemical polishing are used [25].

For decorative purposes, anodized aluminum can be colorized by using organic or inorganic dyes. The dye infiltrate the pores of the oxide layer, giving the color of the surface. The colorizing of the anodized aluminum related to the precipitation of insoluble inorganic salts in the pores, hereinafter referred to as two-step colorizing, occurs according to an example reaction:(3)$${\left({\mathrm{CH}}_{3}\mathrm{COO}\right)}_{2}{\mathrm{Pb}}_{\mathrm{solution}}+{\left({\mathrm{NH}}_{4}\right)}_{2}{\mathrm{CrO}}_{4}{}_{\mathrm{solution}}\to {\mathrm{PbCrO}}_{4}{↓}_{\mathrm{solid}}+2{\mathrm{CH}}_{3}{\mathrm{COONH}}_{4}{}_{\mathrm{solution}}$$

Sealing is a process to eliminate open pores of the anodized aluminum. It is also the process by which the dyes or insoluble salts contained in the pores become stuck as a result of the closure of the pores. During sealing, the oxide layer changes its structure; thus, hydrated Al_2_O_3_·H_2_O (bemite) is formed. This reaction is accompanied by swelling (due to the addition of water of crystallization and changes in the structure of the crystal lattice), which leads to the closure of the pores and the creation of a smooth surface.

Various aluminum alloys are anodized differently [26]. The 6000 alloy series, with the high levels of magnesium and silicon, which influence the anodizing process, are suitable for anodizing and coloring, with a very homogeneous oxide layer. As described in [27], a satin finish causes the surface to become dull. The oxide layer was over 30 microns thick and some parts of the surface were dark gray. For the tests in [27], samples provided by COLOR METAL SRL though SC ANOROM SRL were used, and the 6000 aluminum alloy series aluminum was 6082 SF02 in the form of extruded bars. Tests were made of anodizing as well as black coloring.

In another study [28], the possibilities of black dyeing, inorganic coloring and electrolytic coloring by sulfuric acid of 6061 series were investigated. For black dyeing, jet black was used as a dye, inorganic coloring was performed in two steps with using cobalt acetate and ammonium sulphide solutions and an electrolytic coloring: sulfuric acid, stannous chloride and phenol sulphonic acid. The influence of surface treatment and sealing method on the structure of the oxide layer, corrosion, porosity, etc., was studied. Regardless of the color method, all the porosity values were very low, which confirms the smoothness of the final surface.

In study [25], a two-step anodizing process was performed, resulting in nanoporous anodized aluminum oxide layers using 6082 aluminum alloy. The aim of this study was to choose appropriate anodizing parameters, such as voltage and temperature, using 0.4 M oxalic acid. The effect of these parameters on the morphological characteristics were investigated. Based on [25], the anodizing in present article was performed under similar technological conditions. It is also known that anodizing with oxalic acid turns the anodized aluminum yellow, which should be taken into account in its further coloring. The yellow color of anodized aluminum means that such aluminum is dedicated to coloring yellow or its related colors: green, orange, etc. therefore, in this article, attempts were made to color the anodized material into orange and yellow.

Xylenol orange is used primarily in spectrophotometric determination of aluminum as a colored reagent [29,30]. In study [31], xylenol orange was tested as a corrosion inhibitor for aluminum in trichloroacetic acid. In study [32], with a variety of dyes, methyl orange was used as a corrosion inhibitor for mechanically pretreated aluminum. Yet another study [33] used methyl orange as a dye adsorbed onto films of anodized aluminum oxide for pH sensing purposes. Films were prepared by anodizing aluminum in oxalic acid. As it turned out, depending on the number of dips and the pH of the solution, the dye adsorbed differently, which resulted in a different color of the aluminum.

Lead chromate (PbCrO_4_) is an inorganic salt (often informally written as a mixture of oxides). Lead chromate may exist as a lemon yellow rhombic form, a reddish yellow monoclinic form, and a scarlet tetragonal form, but only the monoclinic form is stable at room temperature. Lead chromes are noted for their excellent opacity, low oil absorption, very bright shades, and high chroma (i.e., they give deep or saturated shades), making them ideal for full shade yellow paints. They also possess excellent solvent resistance and moderately good heat stability, which can be further improved by chemical stabilization [34]. Lead chromate pigments, whose color index CI Pigment Yellow is 34, a popular yellow dye [34] traditionally used for pipe and cable applications, are linked with the phase out of lead-based stabilizers and automotive paints [35].

Although methods for modifying the aluminum surface by anodizing and further processing are known [23,24,27,28,32,33,36,37], such tests have not been performed on recycled alloy 6082. This article shows practical examples of surface treatment of 6082 alloy: solid and its scrap. Coloring with xylenol orange, noticed for the first time, is shown. Innovative color analysis was performed digitally based on color histograms.

The goal of this research was also to study the differences in the quality of the Al6082 alloy: solid, with coarse and fine chips after chemical surface treatment. For this purpose, several exemplary types of chemical surface treatments were carried out: 1. etching, 2. chemical polishing (shining) by 2a. currentless and 2b. current process, 3. anodizing with using oxalic acid and 4. organic and inorganic colorizing, including 4a. one-step and 4b. two-step processes.

## 2. Materials and Methods

The research material was 6082 alloy in the form of a profile after extrusion with a diameter of 40 mm and a height of 1000 mm. The chemical composition of the initial material and its mechanical properties are presented in Table 1 and Table 2.

The 6082 alloys with different morphology were produced using machining (milling and turning). The machining process was carried out in laboratory conditions without the use of coolant, with the cleanliness of the workstations in the process. The first material used for the tests were chips produced on the ROBGRAF 1 milling machine (Markus–Texi, Poland), and for the purposes of the tests, these chips were marked as fine chips. The chip fraction measured using the imageJ software (1.35j) [39] was 0.315–0.4 mm (Figure 1A). In the milling process, the spindle penetrated the material to a depth of 0.5 mm and moved along the sample surface with a spiral direction, starting from the center and ending with the outer edge. The milling process was carried out at a rotational speed of 28,000 rpm. Another type of material used for the research was the chip, which for the purposes of the research was called coarse chips, the average size of which was: 20 × 5 × 1.8 mm (Figure 1B), these chips were produced in the process of turning on a TUM 35 lathe (Famot, Pleszew, Poland) with a rotational speed of 300 rpm, and a feed rate of 0.2 mm/s.

The chips obtained these ways were the input material for the cold compaction process without a protective atmosphere. Each of the batches weighing 25 g was individually placed in a special container and, using a stamp, was pre-pressed on a PS Logistics 100 presser under a pressure of 240 MPa, which resulted in moldings with a diameter of 38 mm and a height of 10 mm (Figure 2). To obtain solid bars intended for further tests, the materials obtained in the previous stage were subjected to the last operation, which was hot extrusion. The input for the process were briquettes/billets consisting of 6 compacts in the case of chips, and in the case of the reference material—a solid ingot with a diameter of 38 mm and a height of 60 mm. Each time, the batch was placed in a recipient heated to 400 °C and heated for 20 min in order to equalize the temperature in the entire volume of the material. It was followed by a co-extrusion process with a speed of 2 mm/s with the use of a rectangular die 3 mm high and 15 mm wide on a hydraulic press.

Samples for metallographic and anodizing tests obtained after the extrusion process were cut with a BP05d electro-erosion machine (Zakład Automatyki Przemysłowej B.P., Konskie, Poland). The dimensions of the samples for coating application were: height—12 mm, width—12 mm, thickness—3 mm. After cutting, they were subjected to grinding on sandpaper grades: 220, 500, 800, 1200, 2000 and 4000, and then polished with diamond paste with a particle size of 3 µm and 1 µm and finished with OPS polishing agent. The above operations were carried out on the Roto-Pol-11 device (Struers, Copenhagen, Denmark). Observations of the microstructure were performed using a Hitachi SU-70 scanning electron microscope (SEM) (Hitachi Ltd., Tokyo, Japan). The tests of mechanical properties by static tensile test were carried out at a speed of 8 × 10^−3^ s^−1^ using the Zwick Roell Z050 testing machine (Zwick/Roell Group, Ulm, Germany); sample dimensions: measuring base—50 mm, measuring base width—10 mm, thickness of the measurement base—3 mm. Density measurements by Archimedes method were made with a laboratory scales XA 120/250.4Y (Radwag, Radom, Poland). Hardness tests were carried out on a Shimadzu HMV-2 T device (Shimadzu Corporation, Kyoto, Japan) using the Vickers method. After surface treatment, the surface quality was investigated at a 4.8 mm gauge length and 80 µm maximum measuring value on a Hommel Tester T1000 profilographometer (Hommelwerke GmbH, Germany). The electrical conductivity was determined with a SigmaTest 2.069 device (Forester Instruments Inc., Pittsburgh, PA, USA). Electrical conductivity was tested at a constant temperature of 20 °C and reported as the mean of 10 measurements. The course of the experiment and the research scheme are shown in Figure 3. Chemical composition tests carried out on the Foundry-Master-Pro 2 device (Ueden, Germany).

### 2.1. Chemical Surface Treatments of Al6082

For rinsing of samples and preparation of aqueous solutions, distilled water with a conductivity 6.10 µS was used. The following chemical reagents were used: sodium hydroxide—p.a., (POCH, Gliwice, Poland); acetone—99.5%, p.a., POCH; sodium chloride—p.a., POCH; sodium carbonate—p., POCH; sodium phosphate dodecahydrate—technical (98%); oxalic acid dihydrate—p. POCH, xylenol orange—p., POCH; lead acetate trihydrate—p., (WARCHEM, Warsaw, Poland); ammonium chromate(VII)—p. (CHEMPUR, Piekary Śląskie, Poland). The POLSONIC SONIC 0.5 ultrasonic bath (Warsaw, Poland) was used to clean the samples, and the POLSONIC SONIC 9 ultrasonic bath was used for the other treatments. RIGOL DP711 DC source (Warsaw, Poland) was used for current polishing and anodizing. During the anodizing, magnetic stirrer LGG LABWARE uniSTIRRER 7 (Meckenheim, Germany) was used. Water bath from WSL POLAND company (Świętochłowice, Poland) was used for sealing. In all cases, drying in the desiccator (calcium chloride as moisture sorbent) took at least 24 h.

#### 2.1.1. Etching and Cleaning

Samples of Al6082 alloy were etched in 1 M sodium hydroxide solution. The samples polished on sandpapers were immersed in the etching solution for 16 s, then washed with a strong stream of running tap water—the whole operation was repeated twice. The samples were finally washed with a stream of distilled water. The samples were then placed in acetone in an ultrasonic bath. Cleaning in acetone took 1 h. Cleaned samples were partially dried on a paper towel and then allowed to dry in a desiccator.

#### 2.1.2. Chemical Currentless Polishing

Etched and cleaned samples of the Al6082 alloy were placed in a solution with the following composition: 100 g/L NaOH, 370 g/L NaCl (supersaturated solution). The currentless polishing was performed at 50 °C in an ultrasonic bath for 5.5 min. The polishing proceeded with the intense evolution of a large amount of gases (hydrogen). Upon completion of the currentless polishing, the samples were removed from the solution and washed in a stream of distilled water. The currentless polished samples were allowed to dry in a desiccator.

#### 2.1.3. Chemical Current Polishing

Applied method of chemical current polishing is similar to the Brytal method [36]. Etched and cleaned samples of the Al6082 alloy were placed in a solution with the following composition: 15% Na_2_CO_3_ and 5% Na_3_PO_4_ (calculated as anhydrous). The samples were connected to a DC source as anodes. The cathodes were stainless steel plates. The current polishing was carried out at the temperature of 80 °C in an ultrasonic bath for 8 min. For each sample, the volume of the solution was 100 mL. The electrical parameters of the current polishing were 12 V and 3 A. After the current polishing was completed, the samples were removed from the solution and washed in a stream of distilled water. The current polished samples were allowed to dry in a desiccator.

#### 2.1.4. Anodizing

Based on [25], the etched and cleaned samples of the Al6082 alloy were placed in a 0.4 M oxalic acid solution. The samples were connected to a DC source as anodes. The cathodes were lead plates with a 1% addition of silver. Anodizing was carried out for 1 h while stirring the solution with a magnetic dipole at a rotation speed of 500 rpm. Magnetic dipole was just below the anode. The electrical parameters of anodizing were 30 V and 3 A. After the anodizing was completed, the samples were removed from the solution and washed in a stream of distilled water. Anodized samples were allowed to dry in a desiccator.

#### 2.1.5. One-Step Colorizing

Anodized samples of the Al6082 alloy were placed in an aqueous solution of xylenol orange at a concentration of 60 g/L. One-step colorizing was performed at the temperature of 40 °C in an ultrasonic bath for 20 min. Upon completion of the one-step colorizing, the samples were removed from the solution and washed in a stream of distilled water. The colorized samples were sealed (as presented at the end of the chapter).

#### 2.1.6. Two-Step Colorizing

Anodized samples of the Al6082 alloy were first placed in a lead acetate solution at a concentration of 15 g/L. After 20 min, the samples were removed from the solution and washed under a stream of distilled water. Then, the samples were immersed in an ammonium chromate (VII) solution at a concentration of 15 g/L. After 20 min, the samples were removed from the solution and washed under a stream of distilled water. Two-step colorizing was performed at 60 °C in an ultrasonic bath for 40 min in total (2 × 20 min). The colorized samples were sealed (as discussed at the next paragraph).

#### 2.1.7. Sealing

The colored samples of the Al6082 alloy were placed over a water bath in a stream of steam. Sealing was carried out at the temperature of 96 °C for 30 min. The sealed samples were washed in a stream of distilled water. The sealed samples were allowed to dry in a desiccator.

#### 2.1.8. Color Analysis

It is not easy to register and compare colors: it depends on the individual predispositions of the observer (physiology and perception), on the conditions of color exposure (lighting, transparency of the medium surrounding the observed object, etc.) and the method of observation (unaided eye/aided eye). In order to accurately register and compare colors, one should choose one appropriate method, which, regardless of the observation conditions, will give unambiguous results, which will enable analysis and the possibility of making correct conclusions. For example, the photos in [27] can be compared inside this article, but cannot be compared with photos from other articles because the conditions under which these photos were taken are unknown (and probably not taken under the same conditions). Thus, to avoid this type of problem, the best solution may be to digitize the colors in the form of, e.g., color histograms. The validity of using color histograms in chemistry and related fields is described in [40]. Comparing color histograms is free from cognitive bias, but is based only on numerical analysis. This can be helpful when it is not possible to show a colorful photo (as for example in [25])—in this case, you can use the color histogram. Since aluminum coloring is primarily used to increase the artistic and functional value by changing the color, it is important to properly show the color of the obtained colored samples.

Color analysis of the Al6082 alloy: solid, with coarse and fine chips after surface treatment was performed based on the photographic images on a black background and hence color histograms were obtained in the Gimp 2.10.22 software [41].

## 3. Results and Discussions

Figure 4A–C shows the microstructures and the distribution maps of elements after the process of extruding flat bars obtained from solid material, coarse and fine chips. Images were taken in the backscatter electrons mode where the number of emitted electrons depends on the atomic number Z. Observation in the BSE mode allows one to visualize the differences in the composition of the sample with different levels of contrast; the map of the chemical elements was provided using EDX. The samples were observed in a transverse to the extrusion direction. In each of the analyzed materials, two types of phases can be distinguished: those containing magnesium and silicon (dark precipitates) and those containing aluminum, iron, manganese and silicon (light precipitates). The maps of the distribution of elements made it possible to identify the presence of given chemical elements in individual phases. Based on the chemical composition and the analysis of the literature [42,43,44], it might be concluded that the bright areas are the Al (FeMn) Si phase. Literature data show that in this type of alloy, depending on the chemical composition, there are phases with a different stoichiometric composition (e.g., Al_9_Mn_3_Si, Al_5_FeSi, Al (FeMn) Si). These phases may also have a different morphology (they may have a columnar structure, polyhedral structure, as well as appearing in the form of the so-called “Chinese script”). Dark precipitation rich in Mg and Si is the Mg_2_Si phase, the presence of which can also be confirmed in various scientific and research articles [44,45,46]. In the case of materials after the milling and turning process, fragmentation of the precipitates is observed in relation to the reference material; this fragmentation occurred as a result of the applied machining (turning/milling) and the hot extrusion process. For a solid material, the average diameter of the Al (FeMn) Si precipitates was 1.5 μm, and for samples obtained from coarse and fine chips, it was 1.2 μm and 0.9 μm, respectively. Microstructural observations of the materials after the extrusion process did not reveal the occurrence of defects inside the materials, which proves the well-chosen conditions of plastic consolidation. The results of the chemical composition presented in Table 3 confirm that as a result of machining (turning and milling) and hot extrusion, the materials were shredded without changing their chemical composition.

The mechanical properties of the extruded profiles were tested in a uniaxial static tensile test. Figure 5 shows the tensile curves for the Al 6082 alloy samples obtained from solid material, coarse chips and fine chips. The solid material has the highest mechanical properties. The tensile strength in this case was 185 MPa and the yield point was 102 MPa. The other two materials reached the tensile strength of 162 MPa (large chips) and 173MPa (fine chips). The highest elongation values of 15% were obtained for the material made of fine chips; the material obtained from a solid ingot and coarse chips slightly deviated from the material made of fine chips and reached elongation of 13.8% and 14.2%, respectively. For comparison, Krolo et al., testing the same alloy, obtained UTS values of 150 MPa for consolidated shredded forms at the process temperature of 400 °C [47], while Tokarski, conducting similar tests, obtained UTS equal to 160 MPa [48]. Reduced strength properties of profiles after plastic consolidation as compared to solid material may be caused by the presence of oxygen layers on the chip surface, which was confirmed by the authors in [49]. They concluded that in order to ensure good strength properties of the chip after plastic consolidation, two conditions should be met: First, the oxide layers must be broken down to allow virgin metal-to-metal contact, and second, the cumulative value of the ratio of the mean stress to the flow stress must be greater than a constant value. In the stretching curves (Figure 5) for all materials, we can see the so-called “Serration”/Portevin Le Chaterlier (PLC) effect. This is the effect of a step, not a continuous change of stress in the material subjected to the static tensile test. The PLC effect is the result of dynamic strain aging (DSO) which is a dynamic interaction between sliding dislocations and free atoms [50]. The PLC effect is classified [51,52] into three types: A, B and C depending on the nature of the temporal–spatial organization of the deformation bands. Type A corresponds to deformations that propagate continuously along the stretching axis (representing isolated plastic waves). Type B means intermittent (time-oscillating) strain propagation (stop-and-go). Finally, type C means a deformation band that appears randomly/stochastically and does not propagate along the sample subjected to stretching [53]. In the case of the tested materials, a similar degree of intensity and frequency of occurrence of stress instability was observed. The conducted observations allow classifying of the deformation instability to the variant A due to relatively small stress drops. The mechanical and physical properties presented in Table 4, such as density, hardness and electrical conductivity, confirm the effective and efficient plastic consolidation. In the case of the obtained results of physical properties, the obtained results are in the range of the measurement error.

The color histogram is a representation by the number and brightness function of the pixels of the photos. By moving to the right or left of the graph, the function applies to the brighter or darker pixels, respectively. The peak heights in the diagrams show the number of pixels corresponding to a given color value. The presented histograms are linear with values in linear space, both for a single color and for a mixture (RGB).

Histograms were constructed on the basis of photographic images on a black background (Figure 6). In all of these images, the material impact on the appearance of the surface is visible by unaided eye. In the images of the material from coarse chips, regardless of the surface treatment, coarse pieces of chip reproduced on its surface may be seen. Moreover, in this case, the surface is slightly darker compared to the corresponding other materials. Both currentless and current polishing resulted in parallel streaks on the surface of the metal. It happened due to the intense gas generation during polishing, where gas bubbles (hydrogen) cavitated and caused mechanical corrosion of aluminum. During one-step coloring, samples of red-orange color were obtained; the resulting color is the effect of xylenol orange. The freshly colored samples were golden in color; however, they turned red-orange after sealing. Coarse and fine samples are more gray (less colored) compared to the solid sample; hence, the conclusion that coarse and fine are less susceptible to dyeing. As for the two-step coloring, it can be said that the color is uniform over the entire surface of the samples. The exception is material from coarse chips, which, depending on the chip position, is locally darker or lighter.

Observations on the macroscopic scale are well represented by the digital way. Through one-step coloring, solid and fine chips materials were the most homogenous in terms of color, as shown by the color histograms (Figure 7). Generally, in the case of the samples colored using xylenol orange, for the average values of 0.259–0.323, the standard deviation ranged from 0.61 to 0.103, which results from the fact that the subsequent color was the resultant of several primary colors. Coarse chip material gave the least red and more yellow surface. After two-step coloring, yellow color of surfaces was expected, according to the color of lead(II) chromate (reaction 3); however, as the color histograms show, a yellowish-green color was obtained. This happened probably because the anodized samples (and coloring was preceded by anodizing) were dark and eventually yellow on the dark grey which looks like green. Nevertheless, it shows that the coloring was successful in this case as well, albeit with a slightly different end result. The average and median values for the solid and fine chips materials are similar, 0.57, 0.510 and 0.516, 0.514, respectively, while for the coarse chips material they are significantly different, i.e., 0.438, 0.439. Visibly, samples of solid and fine chips materials have a more similar color to each other than to coarse chips material. The color of the coarse chips material is closer to yellow (mean and median values are closer to yellow).

Figure 8 shows microscopic photos of the samples after surface treatment; the images were taken using the SE detector. The polishing samples show fine light particles which do not give off chemical polishing. Based on Figure 4, it is known that these phases are rich in iron, magnesium, manganese and silicon. Both during currentless and current polishing, craters can be observed on the surface, but in current polished samples, there are more of them. These craters were created as a result of cavitation—escaping gases caused mechanical corrosion. To avoid this phenomenon, the polishing process should be carried out at lower values of parameters. The roughness coefficient values were the largest of the remaining samples and ranged from 0.29–1.06; hence, it can be concluded that polishing in the given parameters was not successful, but taking into account the visual aspect, the polishing contributes to a significant brightening of the surface of the samples. Roughness for anodized and colored samples was low, about 0.1, and coloring and sealing did not affect the roughness of the samples. Probably this is due to the sealing, which contributes to the smoothing of the colored oxide film. Anodized sample is characterized by a developed surface; hence, it appears darker on a macroscopic scale compared to raw aluminum. One-step coloring of microscopic photos do not differ much from photos of anodized samples. The red dye, which is xylenol orange, does not change the surface microstructure, but only its macroscopic color. It is different in the case of two-step coloring. Characteristic needles appear in the microstructure. These needles are lead-chromium oxide (similar to [54] or [37], in which lead-chromium oxide was crocoite in the form of longitudinal crystallites), which gives the surface a greenish-yellowish color.

From the photos of the cross-sections of the samples near the surface (Figure 9, the images were taken using the BSE detector), the roughness of the currentless and current polished samples can be seen: for the current polishing, the roughness is greater. The anodized and colored samples show a characteristic barrier layer, which is aluminum oxide. This is a confirmation of the effective formation of the oxide layer during anodizing. This layer is compact, with a thickness of approximately 5–6 microns (Table 5). Its chemical composition, which is presented in Table 5, is approximate to the 1 Al: 1 O relationship (for anodized samples: 0.90–1.06), but the ratio slightly decreases for colored samples (0.85–1.03). A few percent of the carbon value is most likely an impurity of aluminum. In the case of samples colored in a one-step process with an organic compound, an increased carbon content is noted compared to the anodized samples; 4.2–4.7 vs. 3.1–4.2, respectively. The colorant increases the carbon content, but not much, i.e., about 1%. However, this confirms that there is a carbon-containing component in the oxide layer. In the case of two-step coloring, PbCrO_4_ needles are bright precipitates in the cross-section. This is also due to the chemical composition of the layer; there is a slight chromium content, 0.10–0.30 wt.%. and a lead content of several percent, 3.40–3.80 wt.%. The theoretical ratio of lead to chromium mass in PbCrO_4_ is about 3.985, while according to the results from the table, for samples colored with precipitation of lead-chromium oxide, the ratio is 12.355–49.286. The reason for this is probably the cementation phenomenon, which causes metallic lead to precipitate out of the solution. However, this phenomenon may be so slight that it does not interfere with the coloring process.

## 4. Conclusions

Based on the presented research results, the following conclusions can be drawn:it was found that using the plastic consolidation method, it is possible to produce a material with a similar density and electrical conductivity from the fragmented fractions to a press made of a solid ingot;using the plastic consolidation method, it is possible to obtain solid rods from waste materials that meet the requirements of mechanical properties set out in the standards [38]. The tensile strength for the material obtained from fine and coarse chips was lower compared to the solid material by 5.41% and 12.44%, respectively;materials after plastic consolidation show increased plasticity compared to the material obtained from a solid ingot with the same chemical composition;the microstructural tests carried out did not reveal any defects inside the analyzed materials, such as cracks or delamination, which is confirmed by appropriately selected parameters of the extrusion process, such as: temperature, speed or the degree of processing;machining treatment processes (milling/turning) and the extrusion process resulted in the fragmentation of brittle intermetallic phases of the Al (FeMn) Si type, which resulted in an increase in elongation. The smallest particle size of Al (FeMn) Si (0.9 µm) was obtained for fine chips;the surface treatment of the recycled 6082 alloy materials is similar to that of the raw material, but on the macroscopic scale, there are differences. The most important difference is the surface quality of coarse chip samples, the surface of which is darker and reflects the shape of the chips;one-step coloring of solid material gives a more intense color, in comparison to the recycled materials. Coloring with the precipitation of lead chromium oxide gives a color more green than yellow, which was originally assumed;polishing of 6082 alloy samples with given parameters causes excessive roughness. Despite this, the surfaces of the samples are bright and aesthetic;on the microscopic scale, the anodized and xylenol orange colored samples do not differ, as the colorant is not visible under the microscope. PbCrO_4_ precipitation coloring causes precipitation of the characteristic needles of this compound;anodizing of 6082 aluminum alloy by using oxalic acid creates a dense oxide layer with a regular thickness of about 5–6 microns.

## Figures and Tables

**Figure 1 materials-14-05066-f001:**
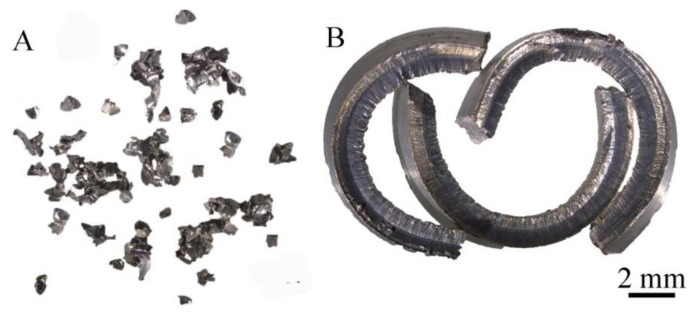
Chips after machining: (**A**) turning, (**B**) milling.

**Figure 2 materials-14-05066-f002:**
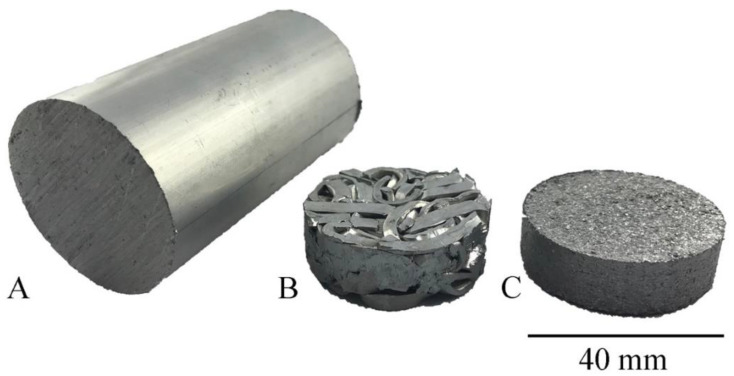
Test material: (**A**) after commercial extrusion, (**B**) after compacting the chip obtained from turning (**C**) after compacting the chip obtained from milling.

**Figure 3 materials-14-05066-f003:**
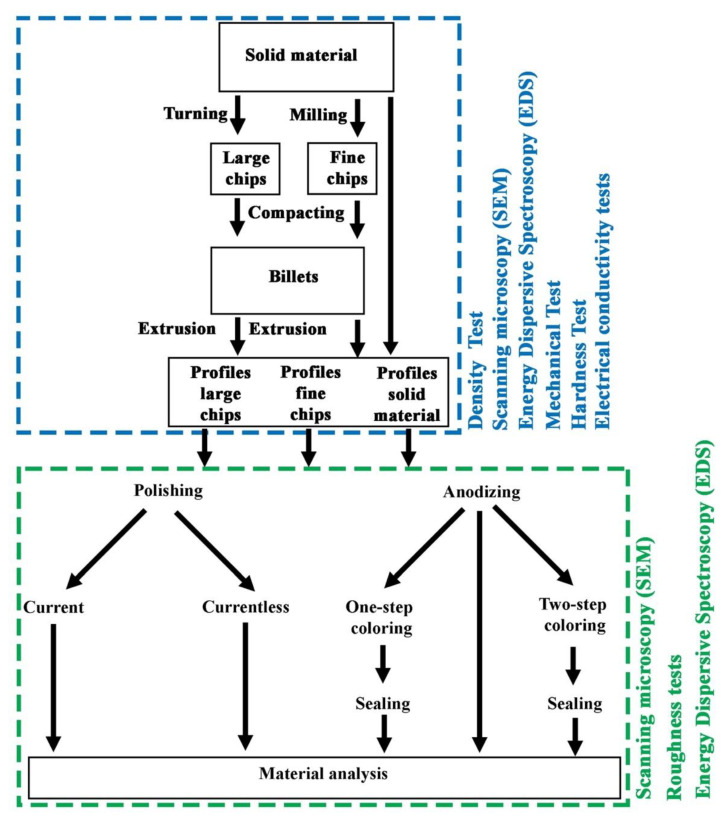
Scheme of experiment and research.

**Figure 4 materials-14-05066-f004:**
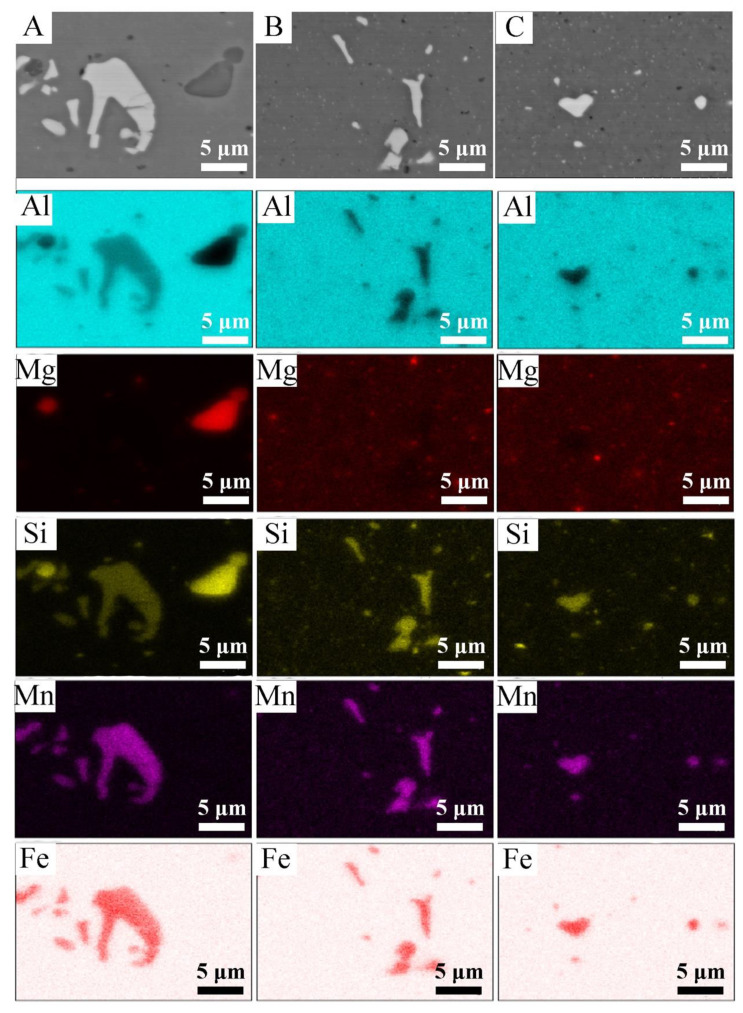
The microstructure in the cross-section of the materials after the extrusion process along with the EDS chemical mapping, (**A**) solid material, (**B**) coarse chips, (**C**) fine chips.

**Figure 5 materials-14-05066-f005:**
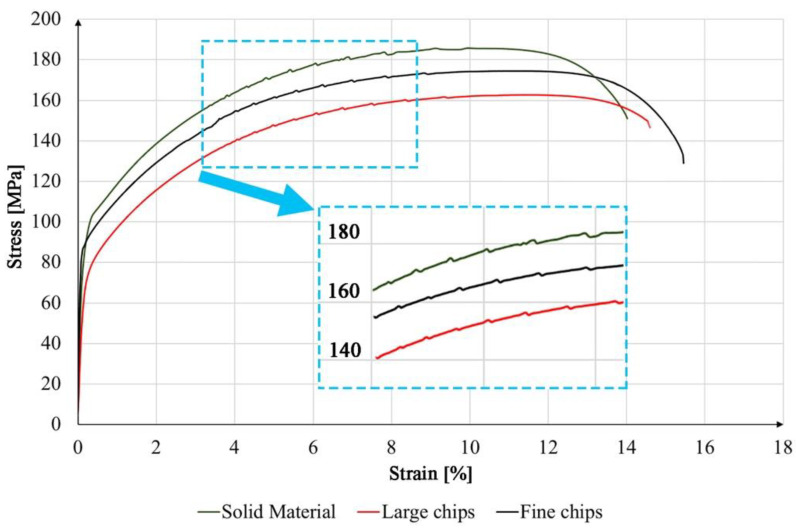
Collective graph of tension curves for solid material, coarse and fine chips.

**Figure 6 materials-14-05066-f006:**
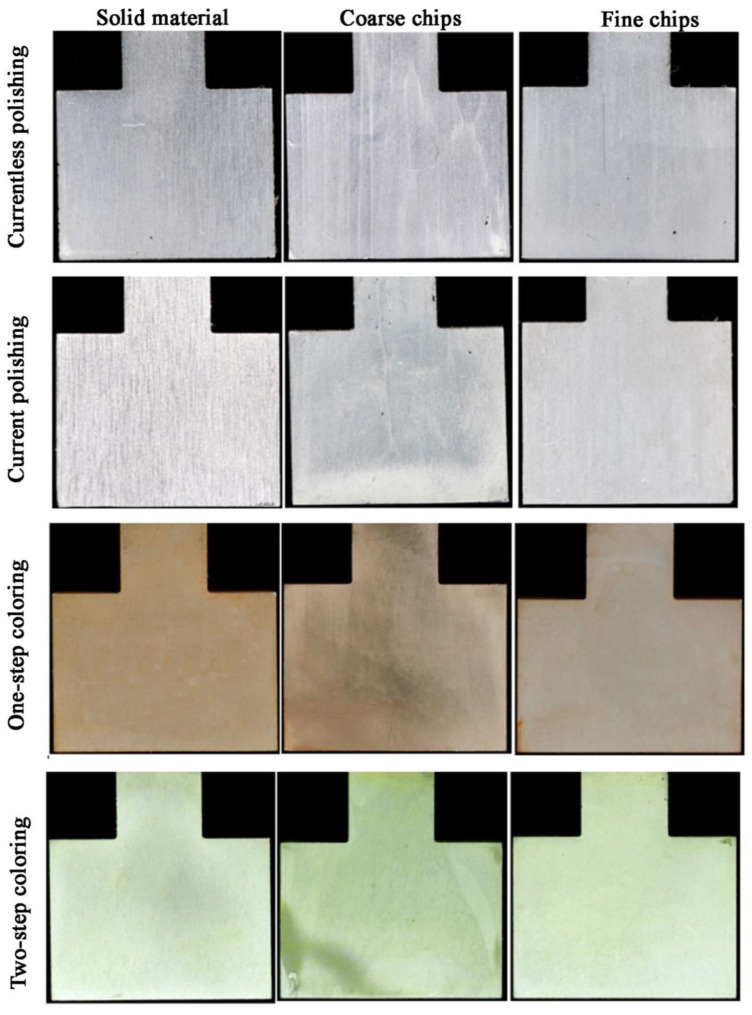
The appearance (unaided eye) of samples after surface treatment of Al6082 alloy: solid, with coarse and fine chips.

**Figure 7 materials-14-05066-f007:**
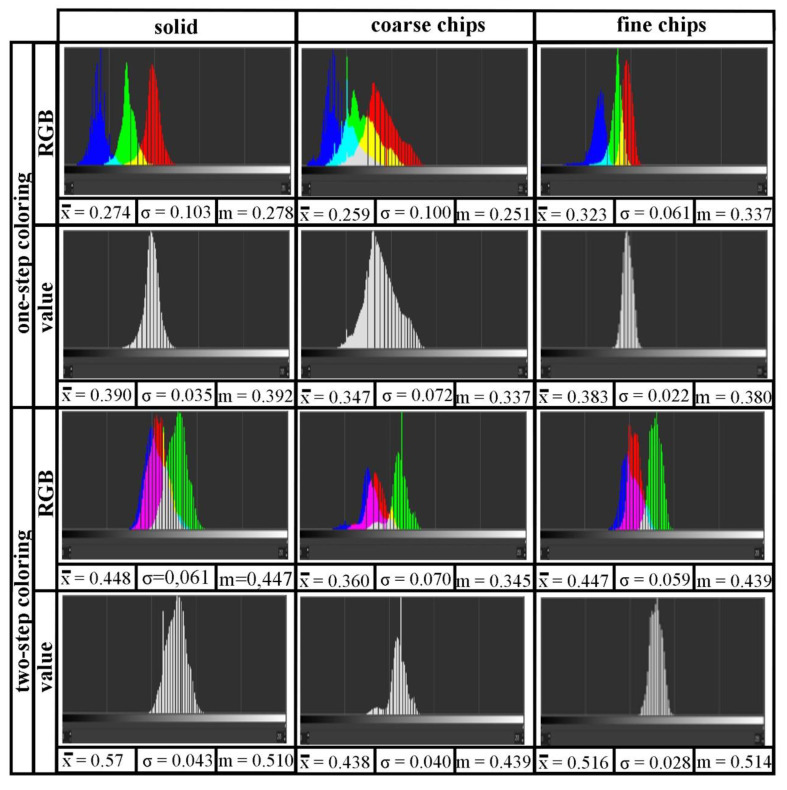
Color histograms of one- and two-step coloring Al6082 alloy: solid, with coarse and fine chips.

**Figure 8 materials-14-05066-f008:**
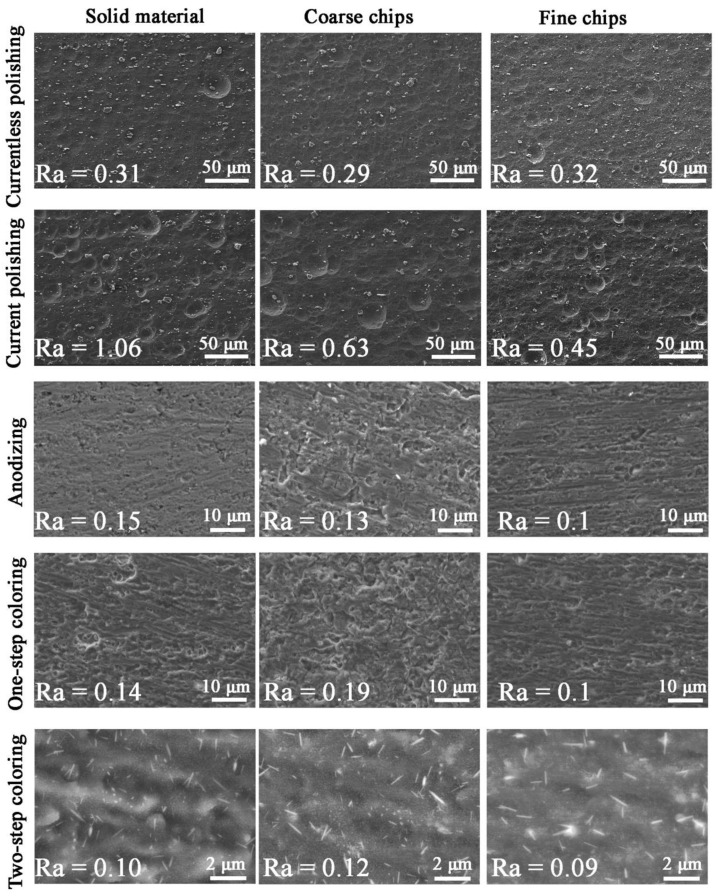
Surface of samples after surface treatment of Al6082 alloy: solid, with coarse and fine chips.

**Figure 9 materials-14-05066-f009:**
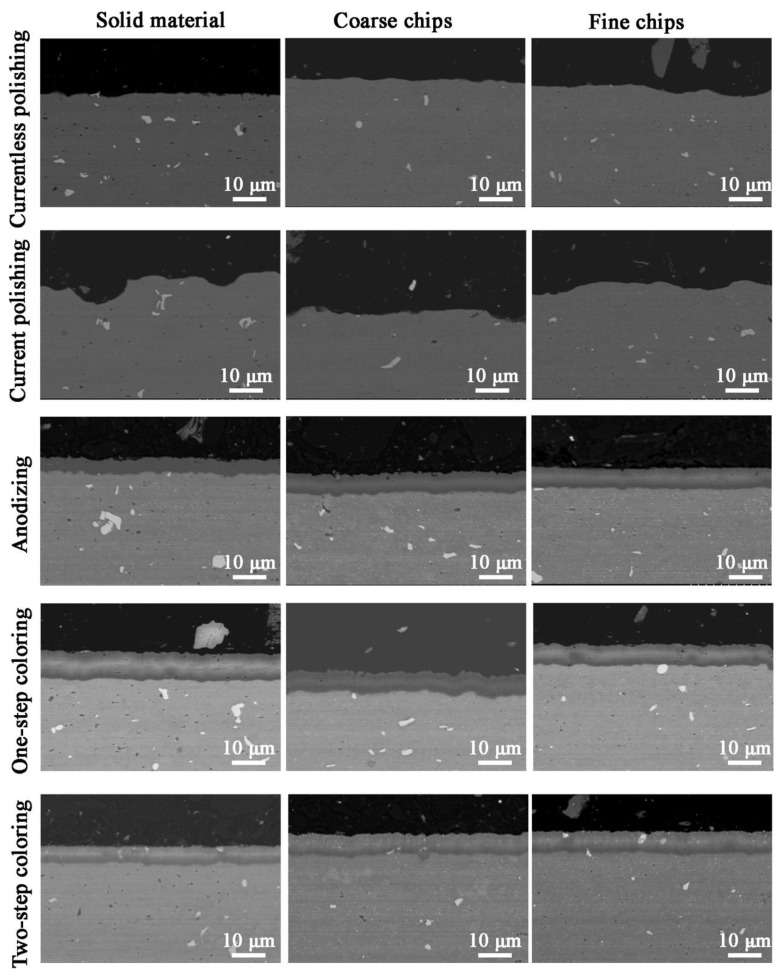
Microstructure of the cross-section of samples after surface treatment of Al6082 alloy: solid, with coarse and fine chips.

**Table 1 materials-14-05066-t001:** Chemical composition of 6082 alloy [38].

Element	Al	Si	Fe	Cu	Mn	Mg	Cr	Zn	Ti	Other
Required by EN 1706 for base metal	95.2–98.3	0.7–1.3	<0.50	<0.1	0.4–1.0	0.6–1.2	<0.25	<0.2	<0.2	<0.15

**Table 2 materials-14-05066-t002:** Mechanical properties of base metal 6082 [38].

Element	UTS, MPa	YS, MPa	Elongation, %	Hardness, HB	Density, g/cm^3^
Required by EN 1706 for base metal	≥150	≥85	≥14	≥40	2.7

**Table 3 materials-14-05066-t003:** Our own spectrometric tests of the chemical composition of solid material, coarse and fine chips.

Materials	Elements									
	Al	Si	Fe	Cu	Mn	Mg	Cr	Zn	Ti	Other
Solid Material	97.23	1.10	0.24	0.03	0.62	0.64	-	-	0.02	0.12
Coarse chips	97.14	1.15	0.22	0.02	0.70	0.69	-	-	0.02	0.06
Fine chips	97.41	1.02	0.21	0.02	0.59	0.64	-	-	0.02	0.09

**Table 4 materials-14-05066-t004:** Mechanical and physical properties of materials after extrusion.

Method	Solid Material	St. Dev.	Coarse Chips	St. Dev.	Fine Chips	St. Dev.
UTS, MPa	185.00	2.03	162.00	1.50	175.00	0.70
YS, MPa	102.00	3.12	72.00	1.70	87.00	1.10
Elongation, %	14.00	2.03	15.00	1.60	15.00	1.20
Density, g/cm^3^	2.70	0.03	2.69	0.03	2.70	0.02
Electrical conductivity, MS/m	29.93	0.05	30.00	0.02	28.93	0.05
Hardness, HV2	52.0	1.1	45.0	1.7	47.0	1.7

**Table 5 materials-14-05066-t005:** The thickness of the oxide layer and the chemical composition.

	Material	Thickness, μm	Standard Deviation, μm	Weight % of Elements				
				Al	O	C	Cr	Pb
Anodized	Solid material	3.965	0.195	49.2	46.6	4.2	-	-
	Coarse chips	4.579	0.159	46.4	50.5	3.1	-	-
	Fine chips	5.788	0.343	45.5	50.7	3.8	-	-
One-step coloring	Solid material	8.125	0.275	45.2	50.1	4.7	-	-
	Coarse chips	5.867	0.152	48.7	47.1	4.2	-	-
	Fine chips	5.845	0.098	45.1	50.3	4.6	-	-
Two-step coloring	Solid material	5.179	0.268	43.6	49.4	3.5	0.10	3.4
	Coarse chips	5.737	0.292	42.9	50.3	2.7	0.30	3.8
	Fine chips	6.131	0.175	42.8	50.0	3.2	0.20	3.8

## Data Availability

All data are provided in full in the results section of this paper.

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
