# Peer review of "Effect of Various Forms of Aluminum 6082 on the Mechanical Properties, Microstructure and Surface Modification of the Profile after Extrusion Process"

_materials, 2021, doi:10.3390/ma14175066_

Round 1

Reviewer 1 Report

Introduction: Please specify the null hypotheses of the study.

Say that anodized aluminum can be colored for decorative purposes. In this study, why color the samples? (line 89)

After anodizing, why do you do the xylenol orange staining? What is the usefulness and necessity of the color analysis? (line 103)

Materials and Methods: Specify the SEM analysis conditions. (line 158).

Figure 3 shows the EDS analysis, but does not appear in the text.

Results and discussion: These two sections must be separated. The authors need to discuss the results in more detail. Additionally, the discussion section must discuss the results obtained with the current literature. 

Conclusions: They are too much long for an article.

Authors must rewrite conclusions that must be shorter and more objective.

Author Response

Thank you very much for taking your time to read our manuscript thoroughly and make valuable recommendations for its correction and improvement. Changes in the article are marked in yellow.

Reviewer 2 Report

The authors studied the mechanical properties, microstructure and surface modification of the compacted and extruded Al 6082 chips and compared them with the solid material.  I recommend a major revision of the manuscript before accepting for publication.

  1. The manuscript is in the lack of references. Such as in L 101: “Although methods for modifying the aluminum surface by anodizing and further processing are known, … “ – some citations should be added.
  2. Where was the solid material supplemented?
  3. In section Materials and Methods please describe firstly the chips after turning (Fig 1A) and then after milling (Fig 1B). The description of the material and corresponding figures should be put in order.
  4. How was the cold compaction process provided? Was it pressed on air?
  5. I do not understand the phrase “This time …” in L 145. Please, correct the sentence.
  6. Figure 4 – what kind of electrons were used to obtain the images Figure 4 A,B,C? Backscatter electrons or secondary electrons? The map of the chemical elements was provided using EDX? Please, specify it also in the section Materials and Methods.
  7. Figure 4 - did the authors prove the presence of possible phases? Such as using TEM?
  8. L 265: What kind of defects did the authors mean?
  9. Did the authors measure the oxygen content of solid material and chips after extrusion?
  10. Why has the solid material better mechanical properties compared to the properties of extruded chips? Please, discuss it.
  11. The authors did not discuss or explain the necessity of the color analysis. The roughness of the surface after surface treatment and the presence of the oxide layer may change the properties of the materials, however, I do not understand the necessity of the color analysis. I recommend to delete the Figure 6 or 7, or I recommend to discuss these results, add some examples and references from other authors. 
  12. The manuscript is overall in the lack of discussion. Please provide more references and discuss the results thoroughly.
  13. The English should be also improved.

Author Response

(The authors gave the same response as above.)

Round 2

Reviewer 2 Report

Thank you for changes and improvement, the manuscript can be accepted in present form.